



# High-resolution mapping of regional traffic emissions by using land-use machine learning models

Xiaomeng Wu[#,1], Daoyuan Yang[#, 2], Jiajun Gu[3], Yifan Wen[1], Shaojun Zhang[1, 4, 5], Rui Wu[2], Renjie
Wang[2], Honglei Xu[2], K. Max Zhang[3], Ye Wu*[,1, 4, 5] and Jiming Hao[1, 4, 5]

[1] School of Environment and State Key Joint Laboratory of Environment Simulation and Pollution Control, Tsinghua University, Beijing 100084, China, Tsinghua University, Beijing 100084, P. R. China

[2] Laboratory of Transport Pollution Control and Monitoring Technology, Transport Planning and Research Institute, Ministry of Transport, Beijing 100028, China

[3] Sibley School of Mechanical and Aerospace Engineering, Cornell University, Ithaca, NY 14853, U.S.A.

[4] State Environmental Protection Key Lab of Sources and Control of Air Pollution Complex, Beijing 100084, P. R. China

[5] Beijing Laboratory of Environmental Frontier Technologies, Beijing 100084, P. R. China

[#] These authors contributed equally to this work.

*Correspondence to*: Ye Wu (ywu@tsinghua.edu.cn)

**Abstract.** On-road vehicle emissions are a major contributor to significant atmospheric pollution in populous metropolitan areas. We developed an hourly-based, link-level emissions inventory of vehicular pollutants using two land-use machine learning methods based on the datasets of road traffic monitoring in the Beijing-Tianjin-Hebei (BTH) region. The results
indicate that a land-use random forest (LURF) model is more capable of predicting traffic profiles than a Gaussian process regression (GPR) model. The inventories under three different traffic scenarios depict a significant temporal and spatial variability in vehicle emissions. One notable finding is that $NO_X$, fine particulate matter ($PM_{2.5}$) and black carbon (BC) emissions from heavy-duty trucks (HDTs) in general have higher emission intensity on the highways connecting to regional ports. Even when traffic restrictions were implemented, a detour of the HDTs in Hebei was observed resulting in relatively
lower emission reductions in Hebei than Beijing. This study demonstrates the power of machine learning approaches to generate data-driven and high-resolution emission inventories, which provides a platform to realize the near real-time process of establishing high-resolution vehicle emission inventories for policy makers to engage in sophisticated traffic management.



## 1. Introduction

The rapid social and economic growth in China has driven the development of road transportation systems and mobility services over the past few decades. This macrotrend also aligns with the faster pace of urban expansion and agglomeration, creating higher travel activities that are not only caused by urban commuting but also are caused by intercity connections. Consequently, on-road transportation systems have resulted in substantial challenges regarding traffic congestion, carbon emissions, air pollution, and land-use issues (Uherek et al. 2010; Waddell 2002; Chapman 2007). To address traffic-related air pollution issues, previous studies have developed link-level emission inventories for metropolitan areas or their urban cores. Notably, more studies have recognized the considerable environmental impact from nonlocally registered vehicles, especially the nonlocal heavy-duty diesel trucks (HDDTs) serving for regional freight purposes. For example, nonlocal HDDTs are estimated to contribute nearly 30% to 40% of the total on-road emissions of nitrogen oxides ($NO_X$) and fine particulate matter ($PM_{2.5}$), which are even greater than the contributions of the 5 million local passenger cars (Wang et al. 2011; Yang et al. 2019). Undoubtedly, for transportation hubs, such as Beijing, we see a clear need to support policymaking that road emission inventories should be enlarged to the multiprovince level to improve the management of transportation emissions.

The technological evolution in intelligent transportation systems has facilitated emission inventories for megacities. For example, Gately et al. applied global positioning system (GPS)-informed speed data from mobile phones and vehicles (Gately et al. 2017) to map the emission fluxes from vehicles in the Greater Boston region. In addition to such trajectory data (Sun, Zhang, and Shen 2018), open-accessed traffic congestion indexes could also be derived from navigation companies or municipal government agencies to dynamically estimate road speeds. For traffic volume and fleet mix, radio-frequency identification (RFID) and traffic cameras are capable of reporting detailed vehicle counts by using license plate numbers (Zhang et al. 2018). These real-world traffic datasets are useful for elucidating temporal and spatial variations in traffic emissions. However, we are still confronted with a few challenges to construct multiprovince, link-level emission inventories by utilizing these developed methods that are applicable to smaller research domains. First, the annual averaged daily traffic (AADT) data, for example, which could be assessed from the U.S. Highway Administration, typically uses the traffic profiles of a select portion of a roadway system (i.e., "sample panel") to represent the "full extent" of the system. Second, simple assumptions and empirical adjustments of vehicle kilometers traveled (VKT) are often used to downscale state-level or national-level AADT profiles to traffic patterns of specific counties (Gately, Hutyra, and Sue Wing 2015). Both of these factors could result in estimates of the spatial variations in traffic volumes that may not represent real-world patterns. Furthermore, the AADT datasets are updated per year according to the annual submission from all states. Therefore, the AADT datasets could support the average analysis of seasonal or day-of-week variations(C. et al. 2014) but are limited to reflecting emissions in a quasi-dynamic fashion (e.g., hourly).

To avoid uncertainty due to simple empirical assumptions, transportation demand modeling has been utilized to assist the development of emission inventories(Gately et al. 2017; Zhang et al. 2018). However, transportation simulation methods are often time-consuming, which has motivated us to explore more efficient data-driven methods to map traffic profiles in an



entire traffic network. That is, machine learning methods can be utilized to analyze the spatial distribution of traffic characteristics relating to some physical land-use features. Compared with traditional parametric methods (e.g., linear land-use regression), machine learning methods are more attractive tools for performing supervised learning tasks on complex datasets by avoiding a prior rigid assumption regarding the nature form of the model. Random forests (RF) (James et al. 2013;

Liaw and Wiener 2001) are often implemented in prediction analyses (e.g., spatial distribution of pollutant concentrations) because of their increased accuracy and resistance to multicollinearity and complex interaction problems compared to linear regression (Hastie 2009; Brokamp et al. 2018). Gaussian processes (GPs) (Williams and Rasmussen 2005) are another machine learning method that could offer flexibility in finding a suitable parametric form for a complex dataset without prior experience. Gaussian process regression (GPR) is a flexible nonparametric Bayesian model that has been successfully applied to predict

traffic characteristics (e.g., traffic congestion(Liu, Yue, and Krishnan 2013) and traffic volumes (Xie et al. 2010)) with state-of-the-art results.

The research domain of this study, the Beijing-Tianjin-Hebei (BTH) region (see Fig. S1), geographically covers three provincial-level administrative regions with a total land area of 217 thousand $km^2$. As the national political center, the BTH region has also developed the busiest freight system in northern China but has suffered from the worst air quality since the

2000s. We utilized hourly traffic profiles including volume, speed and fleet mix obtained from the governmental intercity highway monitoring network to pioneer land-use machine learning methods for developing link-level emission inventories. The methodology can potentially be used either to map traffic characteristics on a larger scale (at the national level) or to deal with real-time urban traffic data streams that need to overcome the challenges of computational accuracy and efficiency.

## 2. Methodology and Data

### 2.1 Research domain and emission calculation

The government traffic monitoring datasets mainly cover main intercity highways, such as expressways, national highways and provincial highways. Notably, we did not include urban sections or minor roads in the research domain because the traffic profiles of these roads were administered by local governmental agencies. The network of intercity main roads in the BTH region has a total length of 50,660 km, including 18,824 km of expressways, 8,989 km of national highways, and 22,847 km

of provincial highways (See the definition of road types in Table S1). The entire region has held 22.38 million registered vehicles (motorcycles excluded) by 2017, with an average annual growth rate of 7% since 2013. In addition, freight trucks for mass transportation of coal and steel from other provinces flood into the BTH region because this region accounts for approximately one quarter of the total steel production in China.

The emissions of primary vehicular pollutants (carbon monoxide, CO; total hydrocarbon (THC); nitrogen oxide, $NO_X$, $PM_{2.5}$,

and black carbon, BC) were calculated with a high-resolution method in a temporal and spatial framework. A link-level emission inventory modeling framework, called EMBEV-Link, was used to complete the emission calculation (Yang et al.



2019). For each road link, hourly emissions are the product of the traffic volume, link length and speed-dependent emission factors (see Eq. 1).

$$E_{h,j,l} = \sum_t \mathrm{EF}_{c,j}(v) \times \mathrm{TV}_{c,h,l} \times L_l \tag{1}$$

where $E_{h,j,l}$ is the total emission of pollutant $j$ on-road link $l$ at hour $h$, units in g h$^{-1}$; $\mathrm{EF}_{c,j}(v)$ is the average emission factor of pollutant $j$ for vehicle category $c$ at speed $v$, units in g km$^{-1}$; $\mathrm{TV}_{c,h,j}$ is the traffic volume of vehicle category $c$ on-road link $l$ at hour $h$, units in veh h$^{-1}$; and $L_l$ is the length of road link $l$, units in km. According to the resolution of traffic mix data, six vehicle categories were defined, namely, light-duty passenger vehicles (LDPVs), medium-duty passenger vehicles (MDPVs), heavy-duty passenger vehicles (HDPVs), light-duty trucks (LDTs), and heavy-duty trucks (HDTs) (see Table S2). Different from the city-scale emission inventories, we did not separate the traffic volumes of transit buses and taxis from the HDVPs and LDPVs, respectively. Additionally, motorcycles were not included because we could barely observe the presence of motorcycles on these intercity highways. For each vehicle category, speed-dependent emission factors were developed based on the EMBEV model. The EMBEV model was developed based on thousands of in-lab dynamometer tests and hundreds of on-road tests (Zhang et al. 2014). Since 2015, this model has become the architype of the official National Emission Inventory Guidebook in China (Wu et al. 2016; Wu et al. 2017). Fig. S2 presents speed-dependent emission factors of CO, NO$_X$ and BC for the LDPV and HDT categories modified by different regions representing the fleet configurations (e.g., fuel type, emission standard and vehicle size) and operating conditions (e.g., fuel quality) estimated for the calendar year of 2017. Evaporative THC emissions were not included in the current EMBEV-Link model because we were limited to spatially specifying the evaporative off-network emissions.

## 2.2 Traffic scenarios under various transportation management schemes

Three traffic scenarios were generated as inputs to the EMBEV-Link model to observe the impact of major transportation management schemes. Scenario Weekday (*S1*) represents the average traffic patterns during weekdays (Monday to Friday) with regular driving patterns. Scenario Weekend (*S2*) represents the average traffic patterns during weekends (Saturday and Sunday), possibly with more leisure travel. Furthermore, Scenario Restriction (*S3*) represents the traffic patterns under special restrictions. The Chinese governments have launched comprehensive actions to alleviate air pollution during seriously polluted episodes. For the on-road sector, transportation restrictions are implemented during polluted days when the PM$_{2.5}$ concentrations or air quality index (AQI) were forecasted to be higher than certain thresholds. As one of the most polluted regions in China, the BTH region has implemented a package of traffic control measures, especially during the winter, with the worst meteorological conditions for pollution dispersion. In this study, Scenario Restriction (*S3*) estimated the real-world traffic patterns during a special week of November 4$^{th}$ to 8$^{th}$, 2017, facing serious haze pollution. More extensive bans than usual were adopted during that week, so that coal-related freight trucks and high-emitting vehicles (e.g., gasoline cars in compliance with pre-China 2 standards) were restricted from many roads in the BTH region. Different from the averaged diurnal patterns in *S1* and *S2*, hourly emissions were continuously estimated throughout the period with traffic restrictions.



## 2.3 Generating dynamic traffic profiles based on land-use machine learning models

**2.3.1 Data collection**

Two land-use machine learning models, namely, the land-use random forest (LURF) and Gaussian process regression (GPR) models, were developed to estimate the spatial distributions of link-level traffic volumes, speeds and mixes for the research domain. An overview of the data used to train the LURF and GPR models is summarized in Table S3 and detailed below.

**Traffic data.** The Ministry of Transport has established nationwide networking to monitor intercity traffic conditions(Yang et
al. 2019; Zhang et al. 2018). Twenty-four-hour diurnal traffic profiles, including the volume, speed and fleet mix, were obtained from 848 intercity highway sites in the BTH region (see Fig. S1). The hourly traffic profiles of a whole week (20[th] to 27[th]) were collected in January, April, July and November in 2017 to represent the average scenarios (*S1* and *S2*). For *S3*, we further collected the hourly profiles from a special week including traffic restrictions (November 5[th] and 6[th]). Fig. S3 shows the distribution of the annual average daily traffic profiles used to train the models to see the capability for predicting the
spatial distribution of these traffic profiles. Notably, the monitoring data of the traffic volumes could not separate the LDPVs and MDPVs from the light-medium-duty passenger vehicles (LMDPVs) on the whole. Therefore, we assembled the two categories when predicting the traffic volumes and separated the predicted traffic volumes according to the estimated total vehicle activity (i.e., registered population $\times$ annual VKT).

**Land-use data.** As land-use machine learning models have rarely been used to develop traffic emission inventories, we
selected candidate spatial predictors based on previous research on the air pollution concentration predictors (Hoek et al. 2008; Lee et al. 2017). Many of these predictors, such as the population density, road density and distance from transportation facilities (e.g., airports and transit stations), are expected to affect traffic activities as well. The potential predictor variables were divided into two groups: a) variables representing a point value and b) variables representing the cumulative values of an area (buffer variables). The buffered variables were represented as a density value (standardized by the buffer area). In total,
139 spatial predictor variables (see Table S3) regarding the land-use data were calculated considering the data availability. Gong et al. transferred the global training sample set developed in 2015 at a 30-m resolution to map a 10-m resolution global land cover in 2017 (Gong et al. 2019), which was assigned into each buffer to calculate the land-use variables. Next, we utilized the points of interest (POIs) mined from the Amap API to calculate the POI density in every buffer and the distance variables (Map 2017). The population density in the buffer was extracted from WorldPop, which estimates the numbers of people per
pixel (ppp) and people per hectare (pph) and adjusts the national totals to match the UN population division with a spatial resolution of 100 m(WorldPop (www.worldpop.org - School of Geography and Environmental Science 2015). The China Digital Road-network Map (CDRM) data developed by NavInfo were used in this study. The road information included the location (latitude and longitude), administration, number of lanes and designed speed limit of the monitoring sites. We categorized the intercity traffic monitoring sites by road type (e.g., expressways, national-level highways and provincial-level
highways) and thus calculated the road density in each buffer.



### 2.3.2 Development and validation of the LURF and GPR models

RF is an ensemble learning method that builds on specifically obtaining the bootstrapped aggregation of several regression trees to predict an outcome. The RF method uses a bootstrap sample for each tree and randomly selects a subset of predictors for testing at each split point in each tree to capture complex interactions and maintain a low bias. The land-use and road information datasets aligned to the prediction road links were used to train the RF regression models consisting of 300 trees and a minimum node size of 5.

The GP is a stochastic process assuming that any finite combination of samples has a multivariate normal distribution. The GP is parameterized by a mean function and a covariance function:

$$f(X) = GP(\mu(X), K(X, X^T))$$

where $f(X)$ denotes the outcome, which is estimated with separate GP models; X is the input of the GP models; and $K$ is the kernel function, which is expressed in terms of a kernel that is appropriate for the modeling task. The kernel is associated with a set of hyperparameters that can be learned directly from the data instead of using cross-validation. The fully Bayesian nonparametric formulation of GPs makes them particularly well suited for modeling uncertainty and noise in observations. In this paper, we apply the squared exponential kernel and exact GPR to estimate the parameters of the GPR model with the training data.

In this study, in the process of model development, we used the annual daily average traffic profiles (i.e., traffic volume by category and speed) as the observed responses of the LURF and GPR models, and the land-use datasets (detailed in Section 2.3.1) were used for the input of the models. We conducted an overall 10-fold cross-validation to evaluate the model performance. The entire dataset, including the hourly traffic datasets and the land-use datasets, was randomly split into 10 groups, with each group containing ~10% of the data. In each cross-validation, nine groups of data were employed as training sets to fit the models and make predictions on the remaining group. This process was repeated 10 times until every group was predicted. The Pearson R (r), root mean squared prediction error (RMSE) and mean absolute prediction error (MAPE) between the model predictions and observations were calculated to evaluate the model performance.

The validation results indicated that the LURF model is more capable of estimating the traffic characteristics (see Section 3.1), which was used to further develop the hourly specific LURF models. The relative importance of the predictors from the trained LURF model shows their prediction ability. For each predictor, we permuted the values of this predictor across every observation in the dataset and measured the increase in the mean standard error (MSE) per permutation. Repeating this process for each predictor, a metric stores the increase in the MSE due to the permutation of out-of-bag observations across each input predictor averaged over all trees in the forest and divided by the standard deviation taken over the trees. The larger this value is, the more important the predictor should be. The metric stores the strength of the relationships between the predictors and projections to indicate the relative importance between various predictors.



## 3. Results and Discussion

### 3.1 The results of the traffic prediction

Table 1 summarizes the statistical metrics used to evaluate the model performance for predicting the daily averaged traffic parameters based on the constructed LURF and GPR models. The performances of the LURF and GPR models were assessed using a 10-fold cross-validation. First, the LURF models consistently derive higher correlation coefficients between the predicted and observed traffic profiles than the GPR models. The Pearson's r values range from 0.62 (LDTs) to 0.79 (LMDPVs) for the LURF models, which are 6% to 36% higher than the corresponding correlation coefficients for the GPR models. Next,

the LURF models derive comparable or lower MAPE values for all traffic profiles than the GPR models. The largest reduction in the mean MAPE values is found for the MDTs (4.22 for LURF vs. 6.67 for GPR). For the other traffic profiles, the results also show lower mean MAPE values for the LMDPVs, LDTs, and HDTs by 12% to 18% when using the LURF model than when using the GPR model. For the HDPVs and speed, the MAPE values are comparable between the LURF and GPR models. In addition, the MAPE distributions depict wider variations when using the GPR models than when using the LURF models.

Furthermore, the RMSE values of the category-resolved traffic volumes by using the LURF models are significantly lower than those for the GPR models by approximately 95% or even larger percentages. Only one exception is that the RMSE for the speed prediction increased from 0.36 km h$^{-1}$ to 5.68 km h$^{-1}$ when using the LURF model compared with the GPR model. We further evaluated the model performance for predicting the hourly averaged traffic profiles under different scenarios. Taking *S1* as an example (see Table S4), the validation results show that the LURF model performs better than the GPR model

in terms of Pearson's R and MAPE, which are similar to the results for predicting daily traffic profiles. However, the GPR model is better than the LURF model at reducing the RMSE when predicting the HDPVs, LDTs, HDTs and speed, mainly because of the greater diurnal variations in these variables.

   For the hourly specific LURF models, we ranked the variable importance in the training process utilizing the method above to construct a variable importance measure and then averaged the hourly ranks for each variable to the final importance index

listed in Table S5. Table S5 illustrates the top 10 important variables for all land-use predictors in our LURF models under *S1*. In general, in addition to the road location (province, city and county), the important input variables for predicting traffic volumes are primarily related to road information, such as the road type, road density, number of lanes and designed speed. Especially for the HDTs and speed simulations, the top 10 important variables are almost all related to road information. The effects of land cover and POI information for estimating traffic characteristics are relatively less important than the role of

road information. For passenger fleets (LMDPVs and HDPVs) and light-duty trucks (LDTs and MDTs), some important inputs related to the land-use datasets (e.g., population, POI information) are indicated by the analysis. Notably, these important land-use variables often represent large buffers (e.g., 2000 m to 5000 m). This effect is because intercity monitoring sites are typically located far away from urban areas, where POI information is more densely available.





## 3.2 Traffic activity characteristics of road networks under various scenarios

Different from urban weekday-weekend patterns (Yang et al. 2019), a higher traffic activity is estimated during weekends (*S2*; 931 million veh km) than on weekdays (*S1;* 841 million km). This data implies that more leisure trips during weekends could be captured by intercity highway monitoring data (see Fig. S4). The lower traffic activity during weekdays is mainly observed in Hebei Province (63%); among all vehicle categories, the LDPVs account for ~70% of the total reduced VKT, followed by HDTs (16%) and LDTs (10%).

We annualized the allocation of the total traffic activity by vehicle category in the BTH region by aggregating the daily patterns under *S1* and *S2* (see Fig. S5). Among all the fleets, the LDPVs account for the largest proportion of the total annual traffic activity (64%), followed by the HDTs (18%) and LDTs (9%). For the HDTs, the fractions of the total traffic activity decrease from Hebei (20%) to Tianjin (18%) and thus to Beijing (11%), probably because of two major reasons. First, the passenger traffic activity by LDPVs is denser in Beijing than in Hebei. Second, many HDTs are strictly limited within the Sixth Ring Road in Beijing, which could also shrink the freight activity outside the restriction area (Yang et al. 2019). Table S6 illustrates the allocations of traffic activity by the vehicle category and road type. For freight transportation, more than 50% of the HDT traffic activity is estimated to occur on expressways. In Tianjin and Hebei, HDTs travel more frequently on expressways than LDTs to improve efficiency.

In terms of temporal variations, the rush hour phenomenon, shown as an increase in the traffic activity, occurred from 8:00 to 10:00 GMT+8 and from 15:00 to 17:00 GMT+8 on weekdays (*S1*). Compared to the rush hour effect within urban areas in Beijing (e.g., peaked at the hour of 7:00 GMT+8)(Yang et al. 2019), the morning rush hours occur later, while evening rush hours are earlier for these intercity highways. Similar trends are observed during weekends (*S2*), with a higher traffic activity reflecting more casual intercity trips (see Fig. S4). In contrast, the diurnal fluctuations of the average speeds depict quite close characteristics between weekdays and weekends (see Fig. S6) because the level of congestion for intercity highways (even during rush hours) is not comparable to urban areas. It should be noted that the speed in Beijing are obviously lower than that in Tianjin and Hebei, mainly due to the congestion occurred in Beijing resulting from its larger travel demands.

The traffic activity of Scenario *S3* with special policy interventions was clearly reduced in the BTH region. The daily traffic activity of *S3* was deceased by 23% compared with that of normal weekdays. However, traffic reductions are heterogeneous among the various vehicle categories and different regions. As Fig. 1 shows, the LDPVs show uniform reductions of approximately 25% on all the intercity highways in the BTH region; only 4% of roads are identified with increased LDPV volumes. For HDTs, reduced traffic could also be observed in Beijing and the expressways and national highways in Tianjin, but there is a certain number of provincial highways in Tianjin (~20%) and expressways (~30%), national highways (~15%) and provincial highways (~20%) in Hebei with increased flow. Four subregions are indicated in Fig. 1 with significantly increased HDT volumes (more than 50%). Subregions 1-3 represent the roads heading to several large ports (the ports of Qingdao, Tianjin, and Qinhuangdao for subregions 1-3, respectively), and subregion 4 represents the areas near the boundary of the BTH region. These differences in traffic volume between the LDPVs and HDTs indicate that for passenger travel, the





restrictions could uniformly reduce the travel demand across the region. In contrast, decreases in HDT traffic volumes are expected in areas with the stricter enforcement of traffic restrictions (i.e., Beijing and highway and national roads in Tianjin).

The traffic restrictions were more effective in Beijing, resulting in a 29% decrease among all vehicle fleets, especially for the HDTs (a 52% decrease) and MDTs (a 42% decrease). In Hebei, such traffic restrictions could result in a detour of the HDTs, as the operators and drivers of the HDTs could conduct their business even on restrictive days. The decrease in the total traffic activity in Hebei was primarily due to the LMDPVs, and the traffic activities of MDTs and HDTs were only 5% and 14% lower than those on normal weekdays (*S1*).

## 3.3 Temporal and spatial characteristics of traffic emissions


The total daily emissions for the intercity highways in the BTH region are estimated to be 1443 tons for CO, 152.3 tons for THC, 1158 tons for $NO_X$, 37.30 tons for $PM_{2.5}$ and 18.73 tons for BC, during weekdays (*S1*, See Fig. 2; see provincial-level total emissions and emission intensity in Fig. S7). During weekends (*S2*), the total daily emissions are estimated to increase by approximately 10% for all pollutants due to increased traffic activity. Comprehensive traffic restrictions under *S3* triggered

decreased vehicle emissions, e.g., 33% to 38% for CO and THC, 23% for $NO_X$, and 15% to 17% for $PM_{2.5}$ and BC in the entire domain relative to *S1*. For LDPVs traveling on the highways, the greater reductions in CO and THC result from the lower traffic volumes due to traffic restrictions, especially in Tianjin and Hebei. In these areas, the controls for vehicles are behind those in Beijing, and therefore, the restrictions on pre-China 2 gasoline cars could result in a larger reduction. Diesel trucks contribute significantly to the emission reductions of $NO_X$, $PM_{2.5}$ and BC, and their lower reduction percentages compared to

CO and THC are related to smaller decreases in the HDT traffic volume in Tianjin and Hebei.

The major temporal difference in the diurnal patterns between *S1* and *S2* lies in the higher emissions during the rush hours of weekends. We thus refer to the weekday scenario (*S1*) to elucidate the temporal and spatial emission patterns (see Figs. 2 and 3). The emission peaks of CO and THC during morning (9:00 to 10:00 GMT+8) and evening (16:00 to 17:00 GMT+8) rush hours, which are apparently associated with diurnal fluctuations in the passenger travel demand, are shown in Fig. S6. As Figs.

4a and 4b visualize, the CO emission intensity close to urban areas is significantly higher than that in outlying areas during both peak and off-peak periods. The highest hourly emissions of CO and THC, which are more associated with passenger traffic activity, were estimated during the morning rush hour (10:00 GMT+8); these emissions were higher than their 24-h averages by approximately 40%-50%. The emission allocations show high resemblance between CO and THC; specifically, the LDPVs dominate the contributions, and the proportions in Beijing and Tianjin are higher than those in Hebei. This increase

is because these two metropolitan areas have a higher density of residential population (indicated by *pop_5000m*), business units (indicated by *POI_office_5000m*), and urban lands (indicated by *urbanland_5000m*) than in Hebei; these variables have been identified as the most important variables for traffic activity predictions of LDPVs by using LURF modeling (see Table S5).

Diesel fleets are responsible for much greater shares of on-road $NO_X$, $PM_{2.5}$ and BC emissions than CO and THC emissions.

Consequently, distinctive traffic behaviors of these diesel fleets will result in disparate temporal and spatial emission patterns





from those for CO and THC. First, we have observed that the total emissions of NO$_X$, PM$_{2.5}$ and BC during the night (0:00 to 4:00 GMT+8, Fig. 2) are closer to the emissions during the daytime, but the nighttime-daytime differences in the emission patterns are less than those of CO and THC. This finding is because a considerable part of long-haul freight trucks in China are operated by two drivers, who could shift duty and travel during nighttime. Second, the NO$_X$, PM$_{2.5}$ and BC emission

contributions by HDTs in Tianjin and Hebei are higher than those in Beijing by approximately 10%. Additionally, the higher percentages of the total NO$_X$, PM$_{2.5}$ and BC emissions by HDPVs (tourist and intercity coaches) in Beijing are higher than those in Tianjin and Hebei. The comparison results indicated higher passenger travel demand in Beijing due to its attraction of tourists and lower freight transportation activity due to truck restrictions.

The emission maps discern Tianjin Port, the largest port in northern China with an annual freight handling amount of 466

million tons in 2018, as a significant hotspot. A large amount of HDTs flood into Tianjin Port (i.e., Subregion 2 in Fig. 1), leading to significantly higher emissions on adjacent highways throughout the day (Figs. 3C and 3D). The daily variation in the NO$_X$ emission intensity in the port area is more obvious than that in Tianjin and the BTH region, with a peak period from 7:00 to 18:00 GMT+8. The hourly average NO$_X$ emission intensity of Tianjin Port is 1.87±0.42 kg km$^{-1}$ h$^{-1}$, which is 47% and 123% higher than the average levels of Tianjin and the BTH region, respectively (see Fig. S8).

**3.4 Discussion**

**An efficient protocol of dynamically modeling hourly based, link-level emissions**. This study provides a universal analytical method for a high-resolution vehicle inventory at a regional scale, especially for regions including many cities suffering from the difficulty of addressing traffic data at a high spatial resolution. Fig. 4 shows the hourly variations in the traffic activities of LDPVs and HDTs and total vehicle emissions by region during a special week when traffic restrictions

were implemented (November 5[th] to 6[th], 2017). During November 2[nd] to 5[th], the traffic activity resembles that during a normal week (e.g., April 20[th] to 27[th], 2017; see Fig. S9). When the traffic restrictions were being implemented during November 5[th] and 6[th], we observed an ~20% reduction in the VKT of LDPVs and HDTs, an ~30% reduction in CO emissions and an ~20% reduction in NO$_X$ emissions, which resembled *S1* and *S3* because of traffic restrictions. Therefore, the high efficiency of the calculation based on the LURF model provides a platform to realize the near real-time process of establishing high-resolution

vehicle emission inventories, and can dynamically support the further evaluation of environmental benefits from traffic policies and management measures.

**Insights on the spatial allocation of traffic emissions for intercity highways**. Addressing the spatial characteristics of traffic profiles based on limited datasets is a significant challenge for establishing high-resolution emission inventories of on-road vehicles. To overcome this barrier, the allocation of the VKT based on-road information is the most typical way to establish

simplified bottom-up inventories. Zheng et al. allocated the VKT of each county based on the same weighing factors considering the vehicle category and road type. Gately et al. used the same allocation of the total VKT considering the differences in weighing factors according to the but regardless of distinguishing the vehicle category. We combined the two different methods to allocate the VKT according to Eq. 2





$$VKT_{l,vc}^{M2} = VKT_{vc}^{M1} \times \frac{TV_{l,vc} \cdot RL_{l,vc}}{\sum_l TV_{l,vc} \cdot RL_{l,vc}} \qquad (2)$$

where $VKT_{vc}^{M1}$ is the total VKT of vehicle category $vc$ of the BTH region calculated in M1, which indicates this study using link-level traffic data based on the machine learning method; $VKT_{l,vc}^{M2}$ is the allocation VKT on-road link $l$ of vehicle category $vc$ in M2, which denotes the allocation method; $TV_{l,vc}$ is the preallocated traffic volume based on its road type and location on-road (Yang et al. 2019) link $l$ of vehicle category $vc$; and $RL_{l,vc}$ is the length of road link $l$. CO and $NO_X$ are discussed, as they represent the gasoline and diesel featured emissions, respectively.

As Fig. 5 illustrates, compared with M1, M2 tends to underestimate the CO emissions on the provincial highways close to urban areas (80% of the provincial highway links are underestimated) and overestimates the expressways and national highways in remote rural areas, especially in Beijing and Hebei (77% and 62% of the expressways are overestimated in Beijing and Hebei, respectively). This miscalculation is because, for remote rural areas without monitoring traffic data, we tend to preallocate volumes according to the road rank, which means expressways and national highways will be allocated more traffic volumes than provincial roads. For $NO_X$, we observe a similar misestimation in the emission distribution of CO; that is,

approximately 70% of provincial highway links are underestimated, and ~70% and 60% of expressways and national highways are overestimated. The long-tailed distributions of the relative difference of CO and $NO_X$ are illustrated in Fig. S10. Overall, the differences between the two methods are not extreme because 79% and 82% of road links' relative differences for CO and $NO_X$ are within ±50%. The analysis indicates that we may use the simplified M2 with the absence of land-use data while

needing to pay attention to the uncertainty in the project-level emission calculations (e.g., port-related areas).

In summary, compared GPR models, LURF models show good cross-validated accuracy and can be useful for the high-resolution prediction of traffic profiles to establish high-resolution vehicle emission inventories. Other machine learning methods exist for predicting the outcome based on land-use information (e.g., neural network approach and supporting vector machine), and additional research in this field is needed to compare and optimize these machine learning methods. On the

other hand, the spatiotemporal dependencies are not as clearly modeled in machine learning as they are in general linear model frameworks, and future work could derive methods for optimizing the predictors used in machine learning models to improve the accuracy of the prediction.

Temporal and spatial patterns of air pollutant emissions from on-road vehicles are of substantial interest because of the associated potential public health impact. The intake fractions of vehicular pollutants are greatest in areas with high vehicle

usage and population density (Marshall et al. 2005). Air quality simulation models with fine-grained input from high-resolution vehicle emission inventories will be valuable for evaluating the potential health benefits from vehicle emission control measures (Ke et al. 2017). However, we are limited to estimates of detailed link-level emissions for urban areas in Tianjin and Hebei due to the data availability (e.g., traffic volume of urban local roads). Currently, as noted in the introduction, there are already link-level emission inventories in large cities with traffic datasets from the ITS, but a strengthening of the portability

of the analytical method is still needed. This study points a promising way to smart management of traffic emissions in various





regions and cities by combining advanced data-driven techniques and multi-source ITS datasets, which will provide policymakers with a better understanding of how air quality impacts regional and local transportation activities.

## 4. Conclusion

This paper developed an hourly-based, link-level emissions inventory of vehicular pollutants using land-use machine learning
methods based on the datasets of road traffic monitoring in the BTH region of China under three traffic scenarios. The methodology can potentially be used either to map traffic characteristics on a larger scale or to deal with real-time urban traffic data streams that need to overcome the challenges of computational accuracy and efficiency. The major findings can be summarized as follows:

(1) The land-use random forest (LURF) model is more capable of predicting traffic profiles than the Gaussian process
regression (GPR) model.

(2) The important input variables for predicting traffic volumes are primarily related to road information, such as the road type, road density, number of lanes and designed speed.

(3) A higher traffic activity is estimated during weekends than on weekdays, and the traffic activity of *Scenario S3* with special policy interventions was deceased by 23% compared with that of normal weekdays.

(4) $NO_X$, $PM_{2.5}$ and BC emissions from HDTs have higher emission intensity on the highways connecting to regional ports. A detour of the HDTs in Hebei was observed resulting in relatively lower emission reductions in Hebei than Beijing even when traffic restrictions were implemented.

## Acknowledgments

This work is supported by the National Key Research and Development Program of China (2017YFC0211100,
2017YFC0212100), the National Natural Science Foundation of China (41977180), and sponsored by Tsinghua-Toyota Joint Research Institute Cross-discipline Program. K. Max Zhang would like to acknowledge the support from the Cornell China Center and the National Science Foundation (NSF) through grant No. 1605407.

## Code/Data availability

The data that support the findings of this study are available from the corresponding author upon reasonable request.

**Author contribution**

Y. Wu and S. Z. conceived the research idea; X. W., D. Y., J. G, and Y. Wen contributed to new analytic tools; D. Y., R. Wu, R. Wang, and H. X. prepared the data; X. W., S.Z., K.M.Z. and Y. Wu provided valuable discussions on modeling development and paper organization; X. W., D. Y., Y. Wu, K. M. Z. and S.Z. wrote the paper with contributions from all authors.



**Competing interests**

The authors declare that they have no conflict of interest.

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



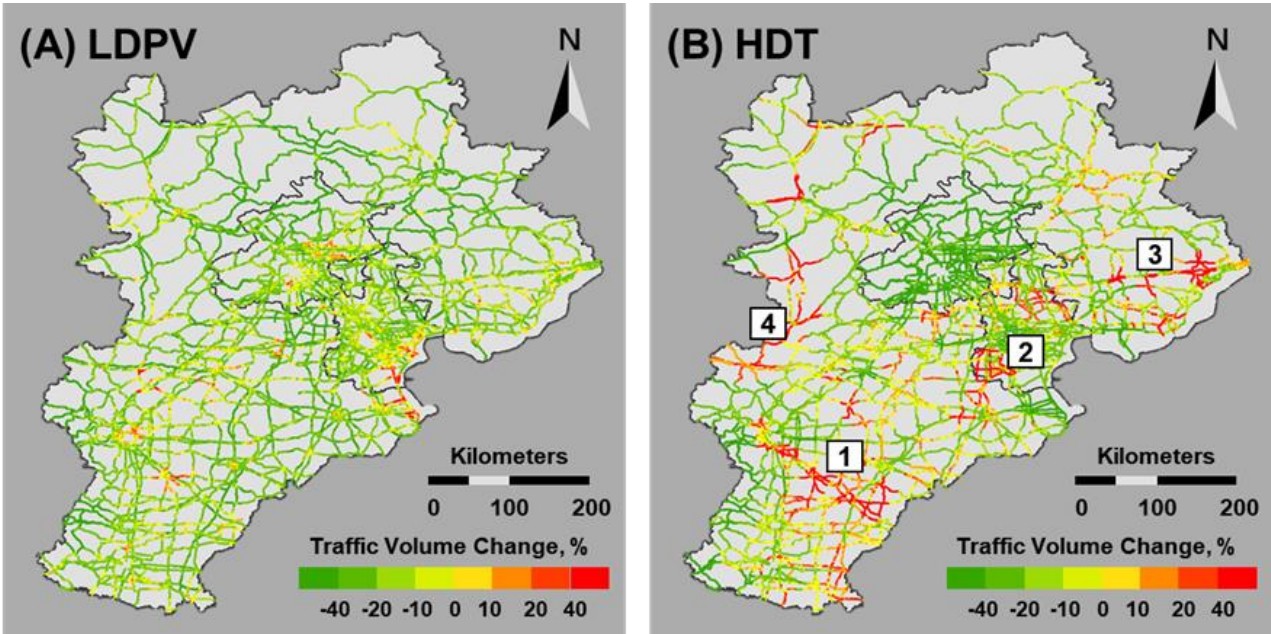

**Figure 1. Estimated traffic volume reductions of the LDPVs and HDTs under traffic restrictions (*S3*) compared to normal weekdays (*S1*).**

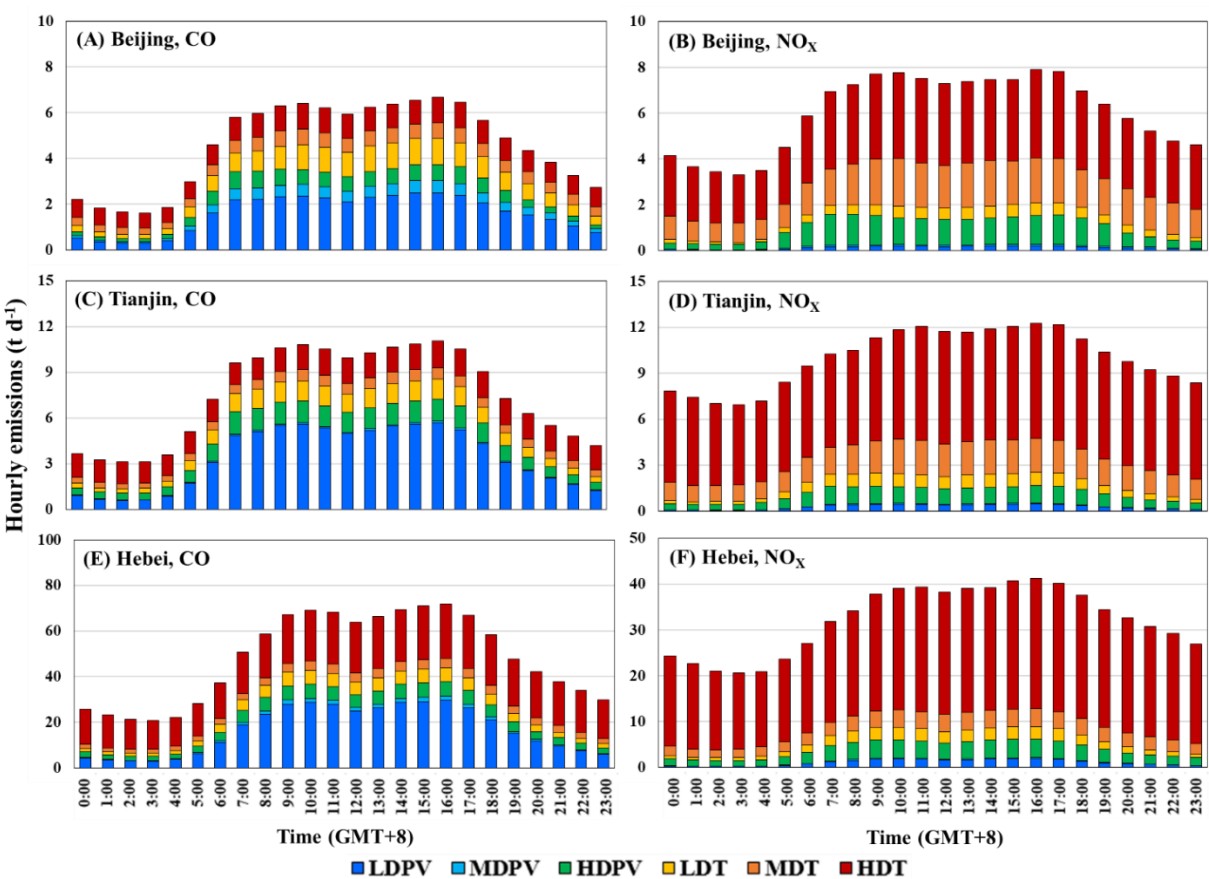

**Figure 2. Estimated hourly emissions by vehicle category under *S1*.**






**Figure 3. Link-based emission intensity of CO (panels A and B) and NOₓ (panels C and D) during a midnight hour (0:00 GMT+8) and a rush hour (10:00 GMT+8).**





**Figure 4. Hourly VKT of LDPV and HDT (A and B) and vehicle emissions of CO and BC (C and D) by region from November 2$^{nd}$ to November 9$^{th,}$ 2017.**

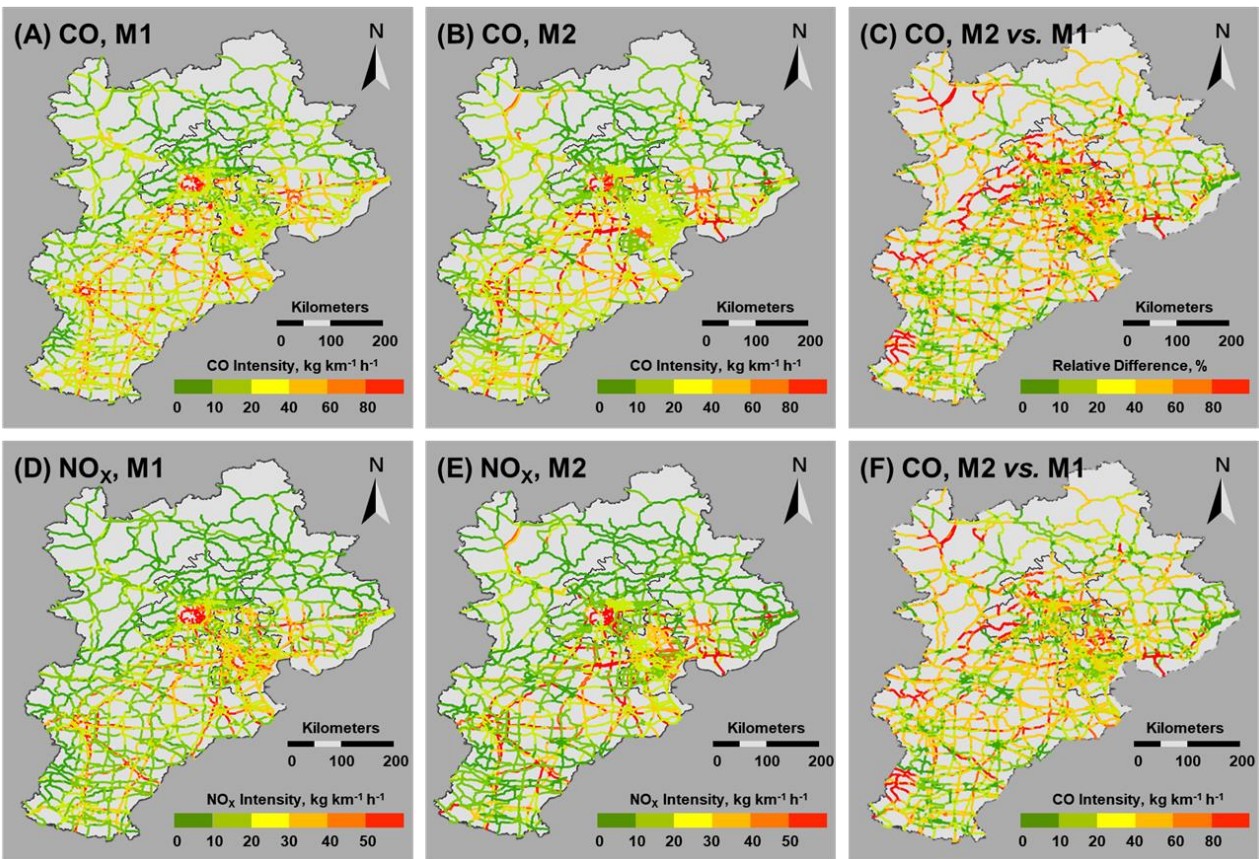

**Figure 5. Comparison of the link-level emission intensity of CO and NOₓ developed by different methods. M1 denotes the emission inventory based on the traffic profiles estimated by LURF models. M2 denotes the emission inventory preallocated based on road type and length, and the total emissions of M2 according to that of M1.**



**Table 1. Cross-validated mean Pearson R (r), mean absolute predictive error (MAPE), and root mean squared error (RMSE) of the LURF and GPR models.**

| Traffic profiles | Pearson's R | | MAPE | | RMSE | |
|---|---|---|---|---|---|---|
| | LURF | GPR | LURF | GPR | LURF | GPR |
| LMDPV | 0.79 | 0.62 | 1.37 | 1.57 | 5360 | 219458 |
| HDPV | 0.61 | 0.46 | 2.92 | 3.10 | 226 | 276419 |
| LDT | 0.62 | 0.49 | 1.26 | 1.41 | 1205 | 30745 |
| MDT | 0.64 | 0.47 | 4.23 | 6.67 | 380 | 16504 |
| HDT | 0.65 | 0.58 | 2.08 | 2.45 | 2706 | 49899 |
| Speed | 0.75 | 0.71 | 0.16 | 0.17 | 5.68 | 0.36 |

Note: The units of RMSE for the traffic volumes and speed are veh d$^{-1}$ and km h$^{-1}$, respectively.