# Peer review of "High-resolution mapping of regional traffic emissions by using landuse machine learning models"

_Atmospheric Chemistry and Physics, 2021_

## Author Comment (AC1)

Reply to comments on "High-resolution mapping of regional traffic emissions by using land-use machine learning models" by Xiaomeng wu et al.

*"Black" means the comments from reviewer and "Blue" text are our responses.*

The reviewers provided very candid and insightful comments on the manuscript. We fully understand that these comments represent the state-of-the-art directions in our research community. We have carefully considered the suggestions of reviewers and tried our best to improve the manuscript. Our responses to the comments are listed below.

Reply to comments from Anonymous Referee #1:

(1) First, I had significant concerns about the scopes of the study. This is a research study with a dominant theme of the transport environment, whereby I could not find any synergies between the scopes of current research and ACP. In terms of technical point of view, I think the research was not designed in an appropriate approach. There are significant confusions in the paper which cause many troubles for the potential readers. The major issue (in my point of view) is the main message of the paper. The paper has designed in two directions of transport and environment. Although the authors tried to provide a new methodology for traffic flow estimations (Transport part) and employ their methodology for emission mitigation strategies (environmental part), neither directions could provide a clear and useful message for potential readers across the world. I would literally suggest them deciding on the direction of the research. They should discuss in detail different machine learning methods, advantages and disadvantages of each method, a comprehensive literature review, why did they select these methods, discuss each of them in detail and conclude the best way for the other parts of the world, if they going to stick with the transport part of their research. On the other hand, they should discuss the existing mitigation strategies, discuss the available literature, ...., and then conclude which scenario is the best and why, if they are going to have the environmental part of their study.

We appreciate it very much for the reviewer's suggestion. First of all, we want to state that the major purpose of this article is to discuss an environmental issue. The Beijing-Tianjin-Hebei region (BTH, study domain of this paper) is one of the most polluted regions in the world according to the global satellite-derived $PM_{2.5}$ pollution profiles. With the rapid clean-up of power plants and industrial sectors, traffic emissions have become an increasingly important source in this region, especially for traffic-populous Beijing. Previous studies were limited to capture the real-world temporal and spatial dynamics of traffic emissions in the BTH region. Therefore, this study starts from the improvement of traffic simulation, combining our previous studies about real-world emissions test and vehicle emission model, to achieve a more precise and efficient simulation of traffic emissions features. In summary, for this study, exploring more accurate traffic simulation is ultimately to achieve more accurate environmental benefit assessment, that is, to build a bridge between transport and environment.

In terms of traffic simulation, we greatly appreciate the reviewer's suggestion on the selection of machine learning methods. We have investigated more machine learning models commonly used in

the environment and transport fields, discussed advantages and disadvantages of each method based on a comprehensive literature review (see Table 1), and further evaluated their applicability in this research. The results showed that the LURF method often performed better than other models in multi-dimensional evaluation indicators (see Table 2). More discussions will be included in the revised manuscript.

Table 1. Advantages and disadvantages of machine learning models used in this study

| Models | Advantages | Disadvantages | Application on predicting traffic |
|---|---|---|---|
| LR | Easy to be applied; Easy to interpret and to be understood | Poor results on non-linear problems due to the linear assumption | To interpret the relationship between traffic variables (Alam, Farid, and Rossetti 2019); Travel time prediction (Zhang and Rice 2003; Rice and Zwet 2004) |
| GPR | Flexible and suitable for a wide range of problems | Low efficiency when solving high-dimensional problems | Dynamic traffic congestion (Liu, Yue, and Krishnan 2013); Short-term traffic volume forecast (Xie et al. 2010) |
| SVR | Works well on non-linear and high-dimensional problems; Perform well on small sample problems | Difficult to choose the optimal kernel; Need to complete feature scaling in advance; Difficult to interpret | Short-term traffic flow prediction (Li and Xu 2021) |
| GBDT | Ensemble learning methods; Able to improve model performance continuously based on the result and the error of last iteration | Easy overfitting; Parameters such as the number of decision trees need to be decided | Traffic volume prediction over a certain time period (Xia and Chen 2017; Yang et al. 2017); Traffic flow prediction considering spatial-temporal relationship (Yang, Zheng, and Sun 2019); Travel time prediction (Li and Bai 2016) |
| LURF | Ensemble learning methods; High computational capacity and high accuracy; Great performance on non-linear and high-dimensional problems; Easy to evaluate the contribution of each independent variable | Easy overfitting; Parameters such as the number of decision trees need to be decided | Road traffic congestion forecast (Liu and Wu 2017) Traffic flow prediction (Gokul L Rajeev et al. 2021) |

Table 2 Simulation performance of machine learning models for traffic prediction in this study

|  | Traffic profiles | LURF | GBDT | SVR | GPR | LR |
|---|---|---|---|---|---|---|
| Pearson's R | LMDPV | 0.79 | 0.81 | 0.65 | 0.62 | 0.48 |
|  | HDPV | 0.61 | 0.54 | 0.51 | 0.46 | 0.3 |
|  | LDT | 0.62 | 0.55 | 0.44 | 0.49 | 0.17 |
|  | MDT | 0.64 | 0.6 | 0.48 | 0.47 | 0.26 |
|  | HDT | 0.65 | 0.58 | 0.56 | 0.58 | 0.5 |
|  | Speed | 0.75 | 0.74 | 0.7 | 0.71 | 0.55 |
| MAPE | LMDPV | 1.37 | 1.37 | 1.25 | 1.57 | 2.06 |
|  | HDPV | 2.92 | 2.85 | 2.64 | 3.1 | 3.05 |
|  | LDT | 1.26 | 1.41 | 1.07 | 1.41 | 1.59 |
|  | MDT | 4.23 | 4.04 | 4.35 | 6.67 | 9.11 |
|  | HDT | 2.08 | 2.24 | 1.81 | 2.45 | 2.71 |
|  | Speed | 0.16 | 0.16 | 0.17 | 0.17 | 0.2 |
| RMSE | LMDPV | 5360 | 7917 | 10715 | 219458 | 13382 |
|  | HDPV | 226 | 536 | 561 | 276419 | 739 |
|  | LDT | 1205 | 1679 | 1741 | 30745 | 2382 |
|  | MDT | 380 | 1024 | 1162 | 16504 | 1546 |
|  | HDT | 2706 | 2207 | 2242 | 49899 | 2596 |
|  | Speed | 5.68 | 10.77 | 11.26 | 0.36 | 15.56 |

(2) This "$NO_X$, $PM_{2.5}$ and BC emissions from HDTs have higher emission intensity on the highways connecting to regional ports." is not a new message for potential readers around the world.

As suggested by the reviewer, we have modified the expression of the conclusions. "Traffic restrictions could result in a detour of the HDTs" might be a more valuable message for potential readers. In addition, we visualized real-world emissions at a large region level which has rarely been reported.

(3) In other words, the results of the present study in this format is a local report and could not be expanded to the other parts of the world or add new values to the scientific committee. As a technical issue, they talked about fleet composition (fleet mix) but they did not mention that how they involve

the role of fleet composition in their emission analysis. Fleet composition is defined as the contribution of vehicle subsets according to their EURO standard (in EU countries), fuel consumption, and/or mileage travelled, etc, to each vehicle class. Fleet composition is totally different from traffic composition (what they report in their paper).

We highly appreciated the reviewer pointed out that there was different between traffic composition and fleet composition. Indeed, we take into account the temporal and spatial characteristics of the traffic composition, which may be not described clearly in the original manuscript. We collected hourly traffic profiles including volume, speed and fleet mix obtained from the governmental intercity highway monitoring network and utilized the data for training machine learning models of traffic network prediction. Therefore, we can obtain traffic composition features in different scenarios, hours and regions (as shown in Fig. 1).

In terms of fleet composition, the EMBEV model (Zhang et al., 2014; Wu et al., 2017) embodies detailed fleet composition of vehicle age, emission standard and fuel type, which has been developed majorly based on registration data. Since the traffic monitoring stations cannot obtain the emission standard information of the vehicle, the proportions of emission standard as well as vehicle age/mileage (used to estimate mileage deterioration of emissions) were assumed to be consistent with the default fleet composition data in the EMBEV model. We also made adjustment based on the restriction policy, such as the HDTs older before China III are not allowed to drive within the fifth rings in Beijing.

Reference:

Alam, Ishteaque, Dewan Md. Farid, and Rosaldo J. F. Rossetti. 2019. "The Prediction of Traffic Flow with Regression Analysis." In, 661-71. Singapore: Springer Singapore.

Gokul L Rajeev, Revathy Nancy, Megha S, Jeena Mary John, and Nithya Elsa John. 2021. 'Traffic Flow Prediction using Random Forest and Bellman Ford for Best Route Detection', *INTERNATIONAL JOURNAL OF ENGINEERING RESEARCH & TECHNOLOGY (IJERT), NCREIS* – 2021 (Volume 09 – Issue 13).

Li, Cong, and Pei Xu. 2021. 'Application on traffic flow prediction of machine learning in intelligent transportation', *Neural Computing and Applications,* 33: 613-24.

Li, X., and R. Bai. 2016. "Freight Vehicle Travel Time Prediction Using Gradient Boosting Regression Tree." *In 2016 15th IEEE International Conference on Machine Learning and Applications (ICMLA),* 1010-15.

Liu, Siyuan, Yisong Yue, and Ramayya Krishnan. 2013. "Adaptive collective routing using gaussian process dynamic congestion models." *In Proceedings of the 19th ACM SIGKDD international conference on Knowledge discovery and data mining,* 704–12. Chicago, Illinois, USA: Association for Computing Machinery.

Liu, Y., and H. Wu. 2017. "Prediction of Road Traffic Congestion Based on Random Forest." *In 2017 10th International Symposium on Computational Intelligence and Design (ISCID),* 361-64.

Rice, J., and E. van Zwet. 2004. 'A simple and effective method for predicting travel times on freeways', *IEEE Transactions on Intelligent Transportation Systems,* 5: 200-07.

Wu, Y., S. Zhang, J. Hao, H. Liu, X. Wu, J. Hu, M. P. Walsh, T. J. Wallington, K. M. Zhang, and S.

Stevanovic. 2017. 'On-road vehicle emissions and their control in China: A review and outlook', *Science of the Total Environmen*t, 574: 332-49.

Xia, Ying, and Jungang Chen. 2017. "Traffic Flow Forecasting Method based on Gradient Boosting Decision Tree." *In Proceedings of the 2017 5th International Conference on Frontiers of Manufacturing Science and Measuring Technology (FMSMT 2017)*, 413-16. Atlantis Press.

Xie, Yuanchang, Kaiguang Zhao, Ying Sun, and Dawei Chen. 2010. 'Gaussian Processes for Short-Term Traffic Volume Forecasting', *Transportation Research Record,* 2165: 69-78.

Yang, Jie, Linjiang Zheng, and Dihua Sun. 2019. "Urban Traffic Flow Prediction Using a Gradient-Boosted Method Considering Dynamic Spatio-Temporal Correlations." *In Knowledge Science, Engineering and Management.* 271-83. Cham: Springer International Publishing.

Yang, Senyan, Jianping Wu, Yiman Du, Yingqi He, and Xu Chen. 2017. 'Ensemble Learning for Short-Term Traffic Prediction Based on Gradient Boosting Machine', *Journal of Sensors,* 2017: 7074143.

Zhang, Shaojun, Ye Wu, Xiaomeng Wu, Mengliang Li, Yunshan Ge, Bin Liang, Yueyun Xu, Yu Zhou, Huan Liu, Lixin Fu, and Jiming Hao. 2014. 'Historic and future trends of vehicle emissions in Beijing, 1998–2020: A policy assessment for the most stringent vehicle emission control program in China', *Atmospheric Environment,* 89: 216-29.

Zhang, Xiaoyan, and John A. Rice. 2003. 'Short-term travel time prediction', *Transportation Research Part C: Emerging Technologies*, 11: 187-210.

128

[Figure]

129

130         Figure 1. Average diurnal fluctuations in hourly traffic activity by vehicle category of the BTH region during various traffic scenarios *S1* to *S3*

Reply to comments from Anonymous Referee #2:

(1) This study established a high-resolution traffic flow database by machine learning methods, but the development of emission factor is not adequately reported. For example, the original EMBEV model was developed for the fleet in Beijing. Please illustrate how to localize the emission factors for the entire fleets in the greater Beijing region.

On the basis of the original EMBEV model, this study updated the BTH emission database, taking full account of the differentiated vehicle emission characteristic of Beijing, Tianjin and Hebei. The main influencing factors include: implementation timetable of vehicle emission standards, fuel quality, intensity of in-use vehicle supervision, proportion of high-emission vehicles, etc. Fig. 2 shows the fleet-average emission factors of CO, NO$_X$ and BC for LDPVs and HDTs estimated by the updated EMBEV model in Beijing, Tianjin and Hebei. The average emission factor in Beijing is lower than Tianjin and Hebei, which is because the control measures for vehicles in Beijing are most stringent and superior. Due to the weak management of in-use vehicles and the higher proportion of high-emission vehicles, the average emission factor of Hebei is the highest.

[Figure]

Figure 2. Fleet-average emission factors for LDPVs and HDTs estimated by the updated EMBEV model.

(2) This paper compared the performance of two machine learning models, LURF and GPR, on traffic flow simulation. The author should explain why these two models are selected.

As suggested by the reviewer, we have investigated more machine learning models commonly used in the environment and transport fields, and evaluated their applicability in this research (see Table 1 and Table 2). More discussions will be included in the revised manuscript. The results showed that

the LURF method used in this study performed better than other models in multi-dimensional evaluation indicators.

(3) Currently, the emission inventory covers a portion of the entire traffic network (highways outside urban areas). Whether the method is applicable to urban roads needs further discussion.

The development in intelligent transportation systems has facilitated emission inventories for urban roads (Gately et al., 2017; Wen et al., 2020). However, the multiprovince emission inventories are established based on empirical allocation by socioeconomic surrogate (e.g., population, GDP) (Zheng et al, 2014; Zheng et al, 2009). This research is designed to improve the efficiency and accuracy of emission inventories on the regional scale, and to construct multiprovince, link-level emission inventories by utilizing developed methods. The recent researches (Yang et al., 2021; Wang et al., 2021) has also showed the inventory calculated based on the LURF model with high efficiency can dynamically support the evaluation of traffic and environmental benefits from traffic policies and management measures in the urban area (i.e., the lockdown during the COVID-19).

(4) Figure 5(F), The title should probably be changed to $NO_x$ instead of CO.

We sincerely thank the reviewer for careful reading. The error has been fixed in our revised manuscript.

Reference:

Gately, C. K. et al., 2017. Urban emissions hotspots: Quantifying vehicle congestion and air pollution using mobile phone GPS data. *Environ. Pollut.*, 229, 496-504.

Wang Y. et al., 2021. Four-Month Changes in Air Quality during and after the COVID-19 Lockdown in Six Megacities in China. *Environ. Sci. Technol. Letters*, 7(11), 802-808.

Wen Y. et al., 2020. Mapping dynamic road emissions for a megacity by using open-access traffic congestion index data. *Appl. Energy*, 260, 114357.

Yang J., et al., 2021. From COVID-19 to future electrification: Assessing traffic impacts on air quality by a machine-learning model. *Proc. Natl. Acad. Sci. U.S.A.* 118(26).

Zheng B., et al., 2014. High-resolution mapping of vehicle emissions in China in 2008. *Atmos. Chem. Phys.*, 14, 9787–9805.

Zheng, J., et al., 2009. Road network-based spatial allocation of on-road mobile source emissions in the Pearl River Delta Region, China, and comparisons with population-based Approach. *J. Air Waste Manage.*,59, 1405–1416.